# ViCrit: A Verifiable Reinforcement Learning Proxy Task for Visual Perception in VLMs

**Xiyao Wang**[1,2†], **Zhengyuan Yang**[2†▽], **Chao Feng**[3,4†]
**Yuhang Zhou**[1], **Xiaoyu Liu**[1], **Yongyuan Liang**[1], **Ming Li**[1], **Ziyi Zang**[5]
**Chung-Ching Lin**[2], **Kevin Lin**[2], **Linjie Li**[2‡], **Furong Huang**[1‡], **Lijuan Wang**[2‡]
[1]University of Maryland, College Park    [2]Microsoft
[3]University of Michigan    [4]Cornell University    [5]Cardiff University
[†]First Authors    [‡]Equal Advising    [▽]Project Lead
xywang@umd.edu    zhengyang@microsoft.com

## Abstract

Reinforcement learning (RL) has shown great effectiveness for fine-tuning large language models (LLMs) using tasks that are challenging yet easily verifiable, such as math reasoning or code generation. However, extending this success to visual perception in vision–language models (VLMs) has been impeded by the scarcity of vision-centric tasks that are simultaneously challenging and unambiguously verifiable. To this end, we introduce **ViCrit** (*Visual Caption Hallucination Critic*), an RL proxy task that trains VLMs to localize a subtle, synthetic visual hallucination injected into paragraphs of human-written image captions. Starting from a 200-word captions, we inject a single, subtle visual description error—altering a few words on objects, attributes, counts, or spatial relations—and task the model to pinpoint the corrupted span given the image and the modified caption. This formulation preserves the full perceptual difficulty while providing a binary, exact-match reward that is easy to compute and unambiguous. Models trained with the **ViCrit Task** exhibit substantial gains across a variety of VL benchmarks. Crucially, the improvements transfer beyond natural-image training data to abstract image reasoning and visual math, showing promises of learning to perceive rather than barely memorizing seen objects. To facilitate evaluation, we further introduce **ViCrit-Bench**, a category-balanced diagnostic benchmark that systematically probes perception errors across diverse image domains and error types. Together, our results demonstrate that fine-grained hallucination criticism is an effective and generalizable objective for enhancing visual perception in VLMs.

## 1   Introduction

Reinforcement learning (RL) has recently emerged as a dominate paradigm [17, 23] for fine-tuning large language models (LLMs) when training tasks are both *challenging* and *automatically verifiable*. Successful examples include mathematical reasoning tasks with concise numerical answers [19, 41], and software engineering problems [78, 38] whose correctness can be checked in a sandboxed environment. By focusing on tasks that strike this balance—sufficiently challenging to have room for improvements yet straightforward to grade deterministically—RL can explore the solution space effectively, extract genuinely useful strategies, and transfer those gains to broader domains.

Despite its success in textual reasoning, RL training with verifiable rewards has yet to demonstrate a comparable significance in improving the visual perception abilities of vision–language models (VLMs). This is largely due to the lack of vision-centric tasks that are both perceptually

39th Conference on Neural Information Processing Systems (NeurIPS 2025).

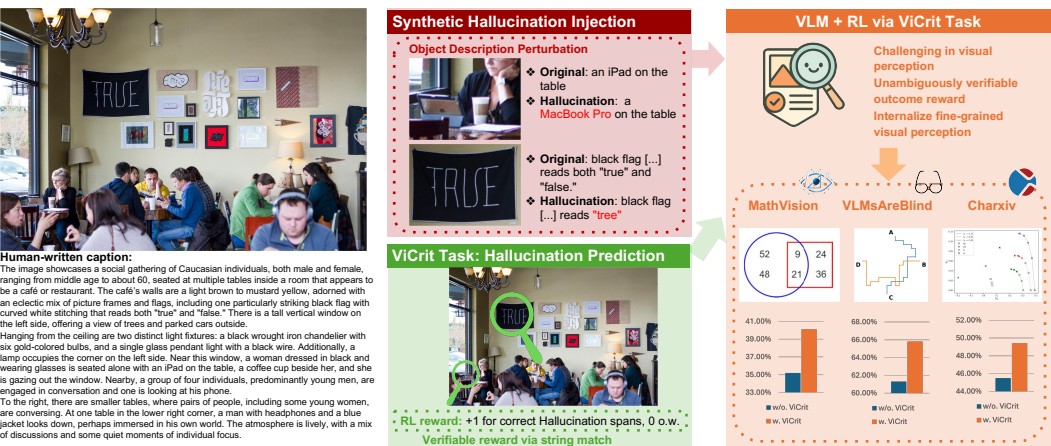

Figure 1: Overview of the ViCrit framework. Starting from high-quality image–caption pairs, we synthetically inject visual hallucinations by minimally altering noun phrases. The model is trained to localize incorrect spans in the caption given the image, receiving a verifiable reward through exact string matching. This fine-grained perceptual objective improves visual perception in vision-language models (VLMs) and generalizes to downstream reasoning tasks across diverse visual domains.

challenging and automatically gradable. Whereas multi-hop math problems naturally compress numerous premises into a single verifiable answer, the semantic elements within an image rarely collapses into such a tidy question-answer pair. Even advanced visual-question-answering benchmarks [1, 21, 51, 35] often probe only fragments of a scene, allowing shallow perception to suffice. Attempts to increase the perceptual difficulty, such as exhaustive image captioning that enumerates every visual element [45, 12, 29, 7, 2, 65], yield paragraph-length outputs (200+ words) that are nearly impossible to grade unambiguously. The central challenge, therefore, is to craft a task that forces the model to perceive the full scene yet produces a concise, deterministically verifiable response.

To bridge this gap, we propose **ViCrit** (*Visual Caption Hallucination Critic*), a reinforcement learning proxy task that offers both *perceptual difficulty* and *evaluation simplicity*. ViCrit trains VLMs to localize synthetic visual hallucinations injected into paragraph-length image captions. It is designed to be both *challenging*, requiring fine-grained visual perception, and *verifiable*, enabling rule-based deterministic reward signals for scalable RL training. As shown in Figure 1, the task begins with human-annotated detailed image captions with more than 200 words [12], and synthetically injects *visual hallucinations*. Such subtle errors misdescribe object, attribute, count, scene text, or spatial relation as its visually similar alternative. The model is trained to act as a critic: given an image and its corrupted caption, it must identify the specific tokens that are incorrect. This token-level span detection can be easily graded via string matching yet requires fine-grained visual perception across the entire image, encouraging the model to internalize robust visual perception strategies extracted during the RL exploration trajectories.

Training Qwen2.5-VL-7B-Instruct and 72B-Instruct with the proposed ViCrit RL task yields consistent gains across ten vision-language benchmarks. In addition to better hallucination benchmark results, these improvements extend well beyond the natural-image domain seen during ViCrit RL training, onto abstract image reasoning and visual math: Qwen2.5-VL-72B-Instruct improves from 35.2% to 40.1% on MathVision [57], from 61.3% to 65.8% on VLMsAreBlind [46], and from 45.5% to 49.4% on Charxiv [63]. These cross-domain improvements indicate that the learned perceptual strategies transfer effectively to general VL domains. By training models to pinpoint fine-grained errors, ViCrit encourages the development of internal perception strategies that cross-check textual claims against visual evidence. Unlike supervised fine-tuning on captioning data [49, 15, 54], which can lead to surface-level memorization, our RL task rewards perceptual correctness and penalizes hallucinations directly. As a result, the model moves beyond merely memorizing the seen object lists, towards learning to decide how to perceive an image. Comprehensive analyses on how ViCrit-induced chain-of-thoughts generalize to a broad spectrum of downstream VLM tasks further reveals the effectiveness and working mechanism of the ViCrit RL training.

In addition to ViCrit training, we present a benchmark named **ViCrit-Bench** for evaluating VLMs on hallucination detection. We group images into four categories and hallucination types into eight fine-

grained hallucination classes, enabling detailed diagnostic analysis. We then manually curate a set of images selected from PixMo-Cap [12] and inject eight types of hallucinations into their corresponding captions. This process results in a high-quality, fine-grained, and highly challenging hallucination detection benchmark, containing 607 samples. The benchmark supports zero-shot evaluation and exposes clear correlations with downstream perception tasks, making it a powerful probe of VLMs' perception limitations. We benchmark a range of state-of-the-art open-source and closed-source vision-language models on **ViCrit-Bench**. Even proprietary systems such as OpenAI-o3 and Gemini-2.5-Pro achieve only 47.7% and 45.2% accuracy. After in-domain reinforcement learning with the ViCrit task, Qwen2.5-VL-72B attains an improved accuracy of 43.0%. The gains are uniform across all four image categories and are especially pronounced on document and abstract images, highlighting the efficacy of ViCrit-based RL for strengthening generalizable visual perception.

Our contributions are summarized as follows:

- We introduce **ViCrit**, an RL task for visual perception that requires VLMs to identify token-level visual hallucinations in paragraph-length image captions. The task is both perceptually challenging and automatically verifiable, enabling scalable RL training with precise, unambiguous supervision.
- Training VLMs with the **ViCrit Task** significantly enhances their performance on a wide range of VL benchmarks. The improvements also generalize to other image domains such as abstract image reasoning and visual math, which shows the advantage of ViCrit incentivizing models to verify visual detail against text, rather than merely memorize seen objects.
- We present **ViCrit-Bench** that systematically probes eight hallucination types across four image domains. The benchmark supports zero-shot evaluation and serves as a diagnostic tool for assessing fine-grained visual perception capabilities in VLMs. Furthermore, its scores track averaged VLM accuracy monotonically, making it a strong indicator of the overall VLM performance.

## 2    Related Work

**Large language model reasoning.** Prompting-based Chain-of-thought methods [64, 25] first explored the reasoning abilities of large language models (LLMs) [5, 10] by eliciting chains of intermediate thoughts, markedly improving arithmetic and commonsense benchmarks [11, 43, 26]. Subsequent decoding strategies aim to further improve test-time performance with extra test-time computation. For example, Self-Consistency sampling [62] that votes over multiple thought paths to boost reliability. Expanding beyond linear traces, structured search frameworks like Tree-of-Thoughts [71] and Graph-of-Thoughts [24] let the model explore a branching space of candidate "thought" states before committing to an answer. Studies [39] also explore hacking the thought process to generate long CoT that is beyond the CoT length distribution. Moving from test-time scaling to training, process reward models [30, 55, 58] grade each reasoning step rather than only final answers, which can be coupled with Monte Carlo Tree Search [67] for fine-grained value estimates. Most recently, large-scale reinforcement learning [17, 23] with outcome-based rewards alone can induce emergent multi-step reasoning skills. The key to its success is challenging tasks that can be automatically verified, such that the RL can be effectively scaled up with minimal noise in its reward signals. The goal for this study is to find such tasks for VLMs' visual perception.

**VLM reasoning.** Based on the modern vision language models [42, 56, 32, 22, 31, 3, 9, 53, 28, 69], recent studies explore the use of multimodal CoT to further improve vision-language reasoning tasks [18, 57, 33, 72] with both grounded textual thoughts [34, 77] and multimodal thoughts [48, 66, 16]. Techniques like rationale distillation and self-reflection further boost these models' reasoning capabilities [76, 70, 81, 59, 68, 61, 14]. Inspired by the success on outcome-based reward based RL in LLMs, recent studies [13, 20, 37, 36, 60, 44, 6, 79, 40] applies similar techniques to visual math and other visual-question-answering benchmarks. Despite the improvements in visual math and STEM questions, they still fall short of significantly advancing fine-grained visual perception.

**Visual-centric VLMs and reasoning.** One threads of works aim to improve visual perception in VLMs via text-supervised visual representation learning [49, 15, 54], which trains the model to generate a good description of the image. This line of work show great promises with the recent success in obtaining ultra-descriptive image captions [45, 12, 29, 7, 2, 4, 65]. However, supervised fine-tuning on captioning data may lead to superficial object memorization, while paragraph captioning task does not have reliable rewards for RL training. In this work, we present an RL proxy task to close this gap.

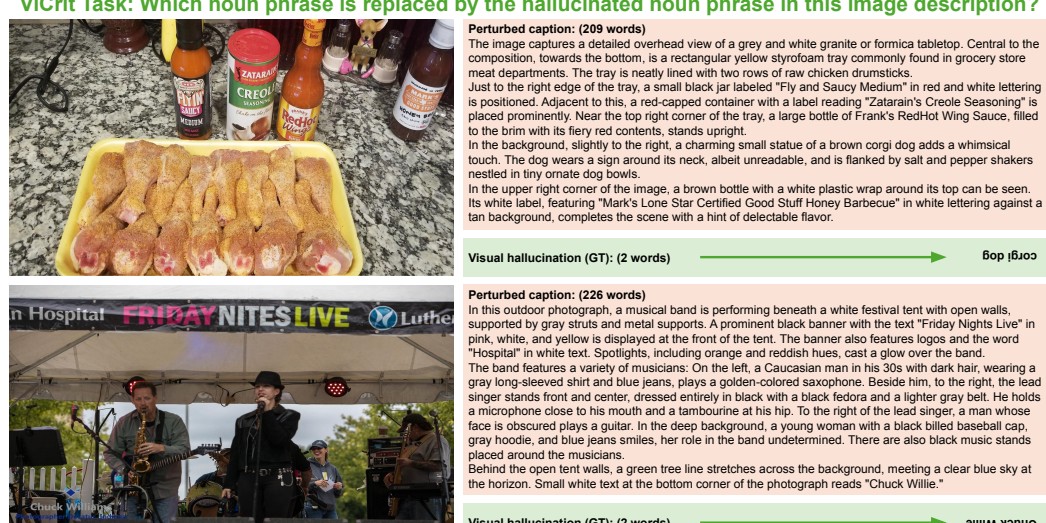

Figure 2: Instead of asking the model to write a paragraph-long caption that is hard to grade (e.g., the 200-word example above), ViCrit feeds the model an almost-correct caption containing a single, deliberately inserted visual hallucination and trains it to locate that error. The short, token-level response is just as demanding in terms of visual perception, yet it is far easier to verify automatically.

## 3 ViCrit RL Training

Recent progress in outcome-based reinforcement learning shows that LLMs learn richer reasoning procedures when trained with *hard* questions whose answers can be *unambiguously* verified. The same recipe, however, is not immediately available to visual perception in VLMs. The traditional caption-supervision objective optimizes a model for recalling a fixed list of objects, but never for deciding where to look next. Our goal is therefore to turn visual perception into an RL problem whose reward (i) compels the model to interrogate every visual details and (ii) remains as cheap and deterministic to evaluate as in code or math.

Examples of our proposed ViCrit task is shown in Figure 2. Instead of asking a model to generate a perfect, paragraph-length caption (200+ words), which is difficult to grade, we present it with an almost-correct caption containing a single, synthetically injected visual error and reward the model for pinpointing the mistaken span (2 words). Solving this task is as hard as perfect captioning: a critic that can reliably spot any hallucination must perceive and understand the entire scene; yet the answer collapses into a few words that can be matched exactly. This simple reshape of the objective delivers the two missing ingredients for perception-centric RL: a genuinely challenging perception task and an evaluation rule that reduces to simple string equality.

### 3.1 ViCrit Task

**Task description.** For every training instance we start with an image $I$ and its exhaustive, human-annotated caption $C$ drawn from the PixMo-Cap dataset [12], with an average caption length of 196 words. We then prompt GPT-4 [42] to select one object description $o$ within that 200-word-length paragraph and perturb it into a visual hallucination $\tilde{o}$, such that the perturbed object is visually similar, semantically plausible, and without ambiguity. We also sample two examples from a small set of manually crafted in-context examples when prompting the LLM. The complete prompt is in Appendix. The desired output is a minimally modified caption $\tilde{C}$ that differs from $C$ by exactly one visual span (*e.g.*, two words in Figure 2). We instruct for diverse types of selected objects $o$ and resulted hallucination $\tilde{o}$, such as object substitution, attribute flip, scene-text error, relation swap, *etc.*.

After data generation we task the model to identity the visual hallucination $\tilde{o}$ given the image $I$ and caption $\tilde{C}$. A positive reward is given if predicted words matches ground-truth $\tilde{o}$. Because the reward depends purely on exact string match, it is deterministic and easy to scale in RL training.

**Discussion on task difficulty.** Perfectly performing the ViCrit task requires the model to perceive the entire visual scene, which is the same level of visual perception demanded of an "oracle" visual captioner that can exhaustively describe every image elements. Indeed, a flawless ViCrit critic could be repurposed into such an oracle by iteratively proposing refinements to an image caption. Thus, ViCrit imposes the same visual perceptual requirements as paragraph captioning, yet its single-span output is easily verifiable, enabling scalable outcome reward based reinforcement learning.

**Data.** We build the image $I$ caption $\tilde{C}$ starting from all samples in the PixMo-Cap dataset [12]. Filtering out the invalid image URLs yields 384K image caption pairs. We then prompt LLM to create the visual hallucination $\tilde{o}$ and use it to replace the original object description $o$ to create the minimally modified caption $\tilde{C}$. In the end, we collect 875K pairs of images and modified captions.

## 3.2 Model Training

We use Qwen2.5-VL as the base VLM for experiments and finetune all model parameters via the ViCrit RL proxy task. We train the model with Group Relative Policy Optimization (GRPO) [50]:

$$\mathcal{L}_{\text{GRPO}} = \mathbb{E}_i\big[\min\big(A_i \cdot \rho_i, \, A_i \cdot \text{clip}(\rho_i, 1-\epsilon, 1+\epsilon)\big)\big], \quad A_i = (r_i - \bar{r}), \quad \rho_i = \frac{\pi_\theta(y_i|x)}{\pi_{\theta_{\text{old}}}(y_i|x)}$$

The sample reward $r_i$ is computed based on a deterministic string matching between the injected visual hallucination string $\tilde{o}$ and the model prediction $\hat{\tilde{o}}$: $r_{\text{answer}} = \begin{cases} +1, & \text{if } \tilde{o} == \hat{\tilde{o}}, \\ 0, & \text{otherwise.} \end{cases}$ We relax the string matching such that the model is not penalized for copying additional words before or after the selected span $o$, as long as they are an exact copy from the original caption $C$. In addition to answer correctness reward, we also follow the standard practice to instruct the model to follow a specific prompt format, which group thoughts with special tokens `<think>...</think>` and final answers with special tokens `\boxed{}`. The format reward $r_{\text{format}}$ is 1 if it correctly uses the special format tokens and 0 otherwise. The final reward for sample $i$ is $r_i = 0.9 * r_{answer} + 0.1 * r_{format}$.

# 4 ViCrit Benchmark

Motivated by the substantial gains yielded by reinforcement learning with the ViCrit task, we hypothesize that zero-shot ViCrit accuracy also correlates with a VLM's perception capability and can therefore anchor a diagnostic benchmark. We thus present **ViCrit-Bench**, a high-quality, fine-grained, and highly challenging benchmark for hallucination detection. In this section, we first present the image domains and hallucination task categories defined in ViCrit-Bench. We then describe the human annotation procedure and dataset construction pipeline. Finally, we provide comprehensive statistics and distributional insights of ViCrit-Bench.

## 4.1 Image Domains and Hallucination Categories

ViCrit-Bench partitions its images into four broad domains, each chosen to probe a complementary slice of visual perception: **(1) Natural images**: everyday photos of landscapes, animals, people, and objects captured in the wild; **(2) Documents**: images dominated by structured content such as tables, charts, plots, diagrams, or dense textual screenshots; **(3) Scene-text–heavy images**: images where scene text is appeared in the scene, such as street signs, memes, comic panels, and illustrative layouts; **(4) Abstract images**: images that do not directly depict real-world objects or scenes, but instead convey meaning through geometric shapes, symbols, color patterns, synthetic compositions, or artistic illustrations; these images emphasize structure, style, or conceptual abstraction rather than natural realism or textual content.

We note that the training data distribution from the PixMo-Cap dataset contains dominated natural images. The annotation on a subset of 4k randomly sampled PixMo-Cap images shows a category percentage of 59%, 10%, 24%, and 7% for image domains 1-4, respectively. Human annotators thus go through additional candidate images to find the proper sources for categories 2 and 4. Based on the image domains above, we then systematically categorized all visual hallucinations into eight distinct hallucination task types, defined as follows: **(1) Count**: evaluates whether the quantity of objects or elements is incorrectly described; **(2) Material**: assesses the model's ability to accurately recognize the material composition of objects; **(3) Spatial**: determines whether the spatial configuration and

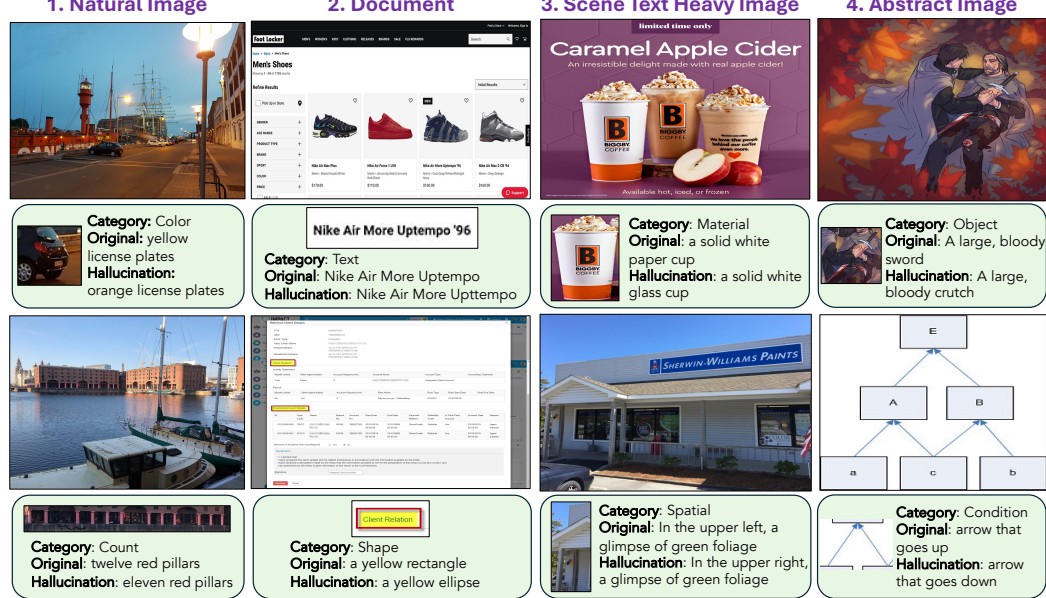

Figure 3: Data examples from ViCrit-Bench, which involve four image categories and eight visual hallucination types. We manually verify each image's long caption, and carefully inject different kinds of proper visual hallucinations.

relative positioning of entities are misrepresented; **(4) Color**: examines the consistency between the described and actual color attributes of visual elements; **(5) Object**: identifies cases where objects are incorrectly classified into wrong semantic categories; **(6) Condition**: checks whether the physical state, dynamic action, or emotional expression of entities is appropriately conveyed; **(7) Shape**: measures the accuracy of describing the geometric structure or contour of objects; **(8) Text**: verifies whether embedded textual content within the image is correctly detected and interpreted.

## 4.2 Annotation Pipeline

All samples in ViCrit-Bench originate from PixMo-Cap, whose 200-word, detailed captions provide a fertile substrate for hallucination synthesis. The construction pipeline proceeds in three stages.

**Stage 1: Image selection and caption sanitization**. For the four image categories, we first employ OpenAI-o3 model to perform an initial classification over the entire PixMo-Cap dataset, identifying candidate images that align with the definitions of natural images, document images, scene-text–heavy images, and abstract images. Subsequently, human annotators manually filter the candidates and select 20 images for each hallucination task with each image category, ensuring images strictly adhere to the domain-specific criteria.

Besides, given that some image captions in PixMo-Cap contain annotation errors, we first used o3 to review and automatically correct the captions, addressing the majority of semantic and factual issues. During the final annotation phase, each caption is individually reviewed and validated by human annotators to guarantee its accuracy and consistency. A total of four human annotators are involved in this process.

**Stage 2: Hallucination injection**. Each image category is assigned to a dedicated annotator, who injects the selected hallucination types by surgically replacing a single noun phrase. This is done by replacing noun phrases with semantically plausible but misleading alternatives that are visually similar, as shown in Figure 3. Each image is allowed to be modified with only one hallucination. For each image category, the number of images per hallucination task is capped at 20. However, in certain categories, some types of hallucinations may be inherently rare or difficult to instantiate—for example, material hallucinations in abstract images—which may result in fewer than 20 finalized examples for those specific tasks.

**Stage 3: Cross-validation**. A final round of cross-validation by two independent annotator ensures the correctness and clarity of the injected hallucinations across all task types.

Table 1: Comparison between **ViCrit-RL-7B** and **ViCrit-RL-72B** with other open-source VLMs. After training on the ViCrit task using Qwen2.5-VL-7B-Instruct and Qwen2.5-VL-72B-Instruct as base models, hallucination rates are significantly reduced, achieving the best performance across all three hallucination benchmarks. Moreover, training on the ViCrit task substantially improves general vision-language performance. On eight general VL benchmarks, ViCrit-RL-72B achieves SOTA results on seven tasks, with the average accuracy increasing from 59.78 to 63.16.

| Model Size | Model | Hallucination Benchmark | | | General benchamrk | | | | | | | | |
|---|---|---|---|---|---|---|---|---|---|---|---|---|---|
| | | CHAIRs ↓ | CHAIRi ↓ | MMHal ↑ | MathVista testmini ↑ | MathVision mini ↑ | MathVerse mini ↑ | MMMU ↑ | MMStar ↑ | MM-Vet ↑ | Blind ↑ | Charxiv reasoning ↑ | Avg. |
| – | GPT-4o | – | – | – | 63.8 | 36.8 | 50.2 | 69.1 | 64.7 | 69.1 | 50.4 | 52.7 | 57.10 |
| | o1 | – | – | – | 73.9 | 58.2 | 57.0 | 78.2 | – | – | 57.0 | 55.1 | – |
| 7B | Molmo-7B-D-0924 | 36.7 | 6.0 | 3.03 | 54.1 | 19.5 | 23.2 | 40.2 | 52.6 | 59.2 | 43.3 | 30.8 | 40.38 |
| | LLaVA-OneVision-7B | 35.0 | 5.5 | 3.12 | 63.2 | 17.4 | 26.2 | 48.8 | 61.7 | 57.5 | 40.1 | 31.3 | 43.28 |
| | InterVL2.5-8B | 29.2 | 5.4 | 3.65 | 64.4 | 22.0 | 39.5 | 54.9 | 62.8 | 68.8 | 47.6 | 32.9 | 49.11 |
| | Qwen2.5-VL-7B-Instruct | 28.0 | 5.1 | 3.74 | 67.8 | 23.6 | 44.5 | 50.6 | 61.7 | 66.0 | 49.3 | 41.4 | 50.61 |
| | ViCrit-RL-7B | 25.2 | 4.5 | 3.77 | 70.7 | 25.7 | 46.3 | 52.0 | 61.9 | 67.1 | 52.6 | 47.8 | 53.01 |
| | Δ (Ours - Qwen2.5-7B) | -2.8 | -0.6 | +0.03 | +2.9 | +2.1 | +1.8 | +1.4 | +0.2 | +1.1 | +3.3 | +6.4 | +2.40 |
| 72B | Molmo-72B-0924 | 28.8 | 5.7 | 3.54 | 61.1 | 24.7 | 30.9 | 48.3 | 58.4 | 65.5 | 46.9 | 35.2 | 46.38 |
| | LLaVA-OneVision-72B | 27.4 | 4.9 | 3.71 | 67.5 | 29.3 | 39.1 | 56.8 | 66.1 | 63.7 | 49.6 | 38.2 | 51.29 |
| | InterVL2.5-78B | 25.9 | 5.2 | 3.89 | 72.3 | 34.9 | 51.7 | **68.7** | 68.9 | 72.3 | 59.8 | 42.4 | 58.75 |
| | Qwen2.5-VL-72B-Instruct | 26.4 | 4.8 | 3.82 | 74.8 | 35.2 | 53.3 | 63.4 | 68.4 | 76.3 | 61.3 | 45.5 | 59.78 |
| | ViCrit-RL-72B | **21.0** | **3.9** | **3.91** | **77.3** | **40.1** | **59.8** | 66.0 | **69.8** | **77.1** | **65.8** | **49.4** | **63.16** |
| | Δ (Ours - Qwen2.5-72B) | -5.4 | -0.9 | +0.09 | +2.5 | +4.9 | +6.5 | +2.6 | +1.4 | +0.8 | +4.5 | +3.9 | +3.38 |

## 4.3 Statistics

Following the aforementioned image selection and hallucination injection procedures, the final ViCrit-Bench contains 607 images, each paired with manually verified and edited captions containing a total of 607 fine-grained hallucination instances. The distribution of hallucination tasks across the dataset is illustrated in Figure 4. All hallucination task types, except for Material, are relatively balanced, each comprising around 13% the total instances. This reflects the comprehensive and well-balanced design of ViCrit-Bench. Due to its unique nature, the Material task appears mostly in first three categories, resulting in a lower overall proportion of 7.9%.

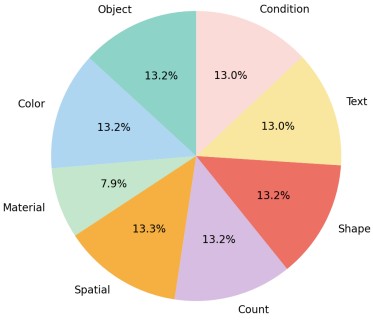

Figure 4: Hallucination task distribution of ViCrit-Bench.

## 4.4 Metric and Evaluation

For each sample, we combine the image $I$ and corrupted caption $\tilde{C}$ with a predefined evaluation prompt template (see Appendix A.2) to form the final evaluation model input. Given this prompt, a VLM must locate the hallucinated span inside $\tilde{C}$ as an open-ended QA task. A prediction is considered correct if the model's prediction $\hat{\tilde{o}}$ exactly matches $\tilde{o}$. We take this string exact match accuracy as the metric for ViCrit-bench.

## 5 Experiments

### 5.1 Effectiveness of ViCrit RL Training

We evaluate the effectiveness of ViCrit-based RL on various VL benchmarks. Through extensive comparisons with SOTA VLMs, we demonstrate the effectiveness of ViCrit as an RL training task and show that reinforcement fine-tuning on this task leads to general VL performance improvements.

**Baseline VLMs.** We start from the Qwen2.5-VL-7B-Instruct and Qwen2.5-VL-72B-Instruct checkpoints. Applying RL training with the ViCrit task produces our ViCrit-RL-7B and ViCrit-RL-72B, respectively. Qwen2.5 models thus constitute our primary models of interest as well as a fairly comparable baselines for ablation. For external comparison, we report benchmark results for three widely used open-source VLMs: Molmo [12], LLaVA-OneVision [27], and InternVL2.5 [9], including both

their 7B-level and 72B-level variants. We also reference proprietary models include GPT-4o and o1. All training and evaluation is conducted with $8 \times 80G$ A100 GPUs.

**Evaluation benchmarks.** *(i) Hallucination mitigation:* we first quantify ViCrit training's impact on dedicated visual hallucination benchmarks. *(ii) Broad VLM generalization:* we then examine if the perceptual skills instilled by ViCrit-based RL transfer to general vision–language benchmarks.

• *(i) Hallucination mitigation.* We adopt two widely used benchmarks: CHAIR [47] and MMHal [52]. Specifically, CHAIR quantifies the proportion of hallucinated content in image captions. Following the setting in previous works [80, 81], we randomly sample 500 images from the COCO Val2014 dataset and use prompts from the LLaVA-150k detailed description dataset, and calculate CHAIR as follows: $\text{CHAIR}_I = \frac{|\{\text{hallucinated objects}\}|}{|\{\text{all mentioned objects}\}|}, \text{CHAIR}_S = \frac{|\{\text{captions with hallucinated objects}\}|}{|\{\text{all captions}\}|}$. MMHal serves as a complementary benchmark for evaluating hallucination in VLMs on VQA tasks. We employ GPT-4 as the scoring model to assess the hallucination severity in model responses.

• *(ii) Broad generalization.* We use 8 widely adopted VLM benchmarks covering mathematical reasoning (MathVista [33], MathVision [57], MathVerse [75]), general knowledge (MMMU [74], MMStar [8], MMVet [72, 73]), visual understanding (Blind [46]), and chart reasoning (ChartXiv [63]).

The middle three columns of Table 1 compares our ViCrit-RL with other VLMs on visual hallucination benchmarks [47, 52]. At the 7B scale, compared with the baseline Qwen2.5-VL-7B, ViCrit-based RL training reduces CHAIRs and CHAIRi from 28.0 and 5.1 to 25.2 and 4.5, respectively. The MMHal increases to 3.77, which surpassing multiple 72B-level models. At the 72B scale, the improvement is even more pronounced: CHAIRs and CHAIRi reaches 21.0 and 3.9, and MMHal improves to 3.91, outperforming all SOTA VLMs across all three hallucination metrics. The consistency of the improvements across scales and benchmarks validates ViCrit's effectiveness in reducing visual hallucination and improving perception across various description types.

Beyond curbing hallucinations, the right side of Table 1 shows that ViCrit-RL consistently lifts accuracy on the eight heterogeneous VLM benchmarks that constitute our "general vision–language" suite. Because the ViCrit proxy forces the model to verify every noun phrase against the image, it refines low-level perception and yields more faithful intermediate representations. These improvements appear to propagate to downstream reasoning tasks, with an averaged improvement of 2.4% on 7B scale and 3.4% on 72B scale. More importantly, the improvements generalize well onto the low-source training image domains. For example, the 72B model improves +4.9% on MathVision, +4.5% on VLMsareBlind and +3.9% on ChartXiv, despite math and abstract images only account for 7% of the PixMo-Cap training data, and 10% for chart and document images. This indicates that the model is not merely memorizing object lists but has learned a transferable strategy for "how to look" at an image before generating text. We provide a qualitative chain-of-thought analysis in Section 5.3 to probe this generalization pattern further.

## 5.2 ViCrit-Bench Results

We benchmark a broad range of SOTA VLMs on our ViCrit-Bench, which probes eight fine-grained visual hallucination types across four image domains. For closed-source models, we evaluate OpenAI-GPT-series which includes 4o, o1 and o3, and Gemini-series which includes 2.0-Flash, 2.5-Flash, and 2.5-Pro. For open-source models, we follow the same experimental setup as Section 5.1 and evaluate Molmo, LLaVA-OneVision, InternVL2.5, and Qwen2.5-VL series. Table 2 shows that ViCrit-Bench is markedly challenging: the best model o3 reaches only 47.7% correctness, while the best open-source model Qwen2.5-VL-72B-Instruct achieves 42.4%.

Spatial hallucination emerges as the dominant failure mode, with the top-performing model achieving only 28.40%, whereas object hallucination and material hallucination looks easier on paper. However, the higher number is because of an easier question subset on foreground objects, and it remains nontrivial to perform perfectly on any one of these eight classes (*cf.* the "corgi" object example in Figure 2.) With respect to image types, "Document Image" and "Abstract Image" are the most challenging ones, as nearly all models exhibited significantly lower accuracy on these two types compared to image types.

Furthermore, RL training with ViCrit task leads to substantial gains on ViCrit-Bench. ViCrit-RL-7B and ViCrit-RL-72B achieves an improved accuracy of 35.6% and 43.0%, respectively. Among four image categories, the largest gains occur on the Document and Abstract image domains, precisely the

Table 2: Evaluation results of a range of closed-source and open-source VLMs on **ViCrit-Bench**. The results indicate that ViCrit-Bench poses a substantial challenge to current models—even the best-performing model, OpenAI-o3, achieves only 47.7 accuracy. After reinforcement fine-tuning on the ViCrit task, ViCrit-RL-72B achieves the highest accuracy of 43.0 over all opensouce models on the benchmark. Moreover, we observe a strong correlation between performance on ViCrit-Bench and the average accuracy on general vision-language tasks for open-source models. Models that score higher on ViCrit-Bench tend to perform better on general benchmarks, suggesting that ViCrit-Bench serves as a reliable indicator of overall reasoning and understanding capabilities.

| Models | General Task Avg. | Overall | Hallucination Type | | | | | | | | Image Type | | | |
| | | | Object | Color | Material | Spatial | Count | Shape | Text | Condition | Natural Image | Document | Scene Text Heavy Image | Abstract Image |
| --- | --- | --- | --- | --- | --- | --- | --- | --- | --- | --- | --- | --- | --- | --- |
| OpenAI-GPT-4o | – | 23.3 | 47.50 | 17.50 | 27.08 | 13.58 | 16.25 | 18.75 | 26.58 | 20.25 | 27.04 | 25.00 | 15.09 | 26.17 |
| OpenAI-o1 | – | 45.8 | 60.00 | 48.10 | 64.58 | 25.93 | 43.75 | 40.51 | **57.69** | 32.91 | **53.46** | 39.29 | 42.14 | **47.26** |
| OpenAI-o3 | – | **47.7** | 67.50 | 46.25 | 60.42 | 22.22 | **50.00** | **62.50** | 54.43 | 22.78 | 51.57 | **49.29** | 43.40 | 46.31 |
| Gemini-2.0-Flash | – | 19.3 | 30.00 | 22.50 | 39.58 | 6.17 | 16.25 | 13.75 | 18.99 | 15.19 | 25.16 | 19.29 | 16.35 | 16.11 |
| Gemini-2.5-Flash | – | 44.4 | 60.00 | 41.25 | 60.42 | **28.40** | 47.50 | 40.00 | 50.63 | 32.91 | 48.43 | 42.86 | 44.65 | 40.94 |
| Gemini-2.5-Pro | – | 45.2 | **68.75** | **50.00** | 66.67 | 20.99 | 43.75 | 50.00 | 39.24 | 30.38 | 46.54 | 46.43 | **52.83** | 34.23 |
| Molmo-7B-D-0924 | 40.48 | 9.6 | 25.00 | 8.75 | 10.42 | 9.88 | 5.00 | 3.75 | 6.33 | 7.59 | 5.66 | 13.57 | 8.81 | 10.74 |
| LLaVA-OneVision-7B | 43.28 | 12.4 | 20.00 | 11.25 | 10.42 | 12.35 | 6.25 | 15.00 | 12.65 | 10.13 | 17.61 | 7.86 | 9.43 | 14.09 |
| InternVL-2.5-8B | 49.11 | 20.0 | 26.25 | 11.25 | 25.00 | 22.22 | 12.50 | 15.00 | 30.38 | 18.99 | 27.04 | 15.00 | 13.21 | 24.16 |
| Qwen-2.5-VL-7B | 50.61 | 21.9 | 30.00 | 23.75 | 39.58 | 9.88 | 12.50 | 8.75 | 45.57 | 12.66 | 35.22 | 12.86 | 20.13 | 18.12 |
| ViCrit-RL-7B | 53.01 | 35.6 | 47.50 | 46.25 | 68.75 | 6.17 | 38.75 | 37.50 | 21.52 | 31.65 | 41.51 | 30.00 | 38.99 | 30.87 |
| Molmo-72B | 46.38 | 18.2 | 36.25 | 11.25 | 10.42 | 11.11 | 6.25 | 13.75 | 25.32 | 27.85 | 23.90 | 18.57 | 13.21 | 16.78 |
| LLaVA-OneVision-72B | 51.29 | 24.5 | 42.50 | 20.00 | 25.00 | 22.22 | 17.50 | 21.25 | 22.78 | 24.05 | 30.82 | 19.29 | 24.53 | 22.82 |
| InternVL-2.5-78B | 58.75 | 32.7 | 46.25 | 21.25 | 45.83 | 22.22 | 20.00 | 26.25 | 46.84 | 37.97 | 38.99 | 29.29 | 31.45 | 30.20 |
| Qwen-2.5-VL-72B | 59.78 | 42.4 | 57.50 | 36.25 | 58.33 | **28.40** | 26.25 | 46.25 | 54.43 | 37.97 | 47.17 | 40.71 | 44.03 | 36.91 |
| ViCrit-RL-72B | 63.16 | 43.0 | 60.00 | 48.75 | **70.83** | 17.28 | 40.00 | 56.25 | 25.32 | **39.24** | 47.80 | 41.43 | 44.65 | 37.58 |

areas where baseline models struggle. This improvement also foreshadows the model's enhanced performance on downstream benchmarks involving multimodal mathematical reasoning and chart understanding after ViCrit-based training, as shown in Table 1.

However, we observe a significant drop in accuracy for the Spatial and Text tasks after RL training. We attribute this to data imbalance in the construction of the training set. Furthermore, RL training led to substantial performance gains for the 7B model on ViCrit-Bench, whereas the improvements for the 72B model are relatively marginal. We hypothesize that this is due to the constructed training set being insufficiently challenging for Qwen-2.5-VL-72B, which already possesses strong visual perception capabilities. To further enhance the performance of the 72B model, more complex and demanding data may be required.

Beyond the raw results, Table 2 exposes a monotonic link between the ViCrit-Bench results and the general VLM performance. Models rank in precisely the same order in the ViCrit-Bench Overall column, as in the averaged VLM performance column quoted from Table 1. Figure 5 demonstrates a strong positive linear correlation between average VLM task performance and ViCrit-Bench scores. ViCrit-Bench scores rise almost linearly with a model's average performance across eight vision-language tasks (r = 0.96), showing that the benchmark effectively evaluates the visual perception, and is a strong proxy for overall VLM capability. In the 7B class, performance rises step-wise from Molmo-7B through LLaVA-OneVision-7B, InternVL 2.5-8B, and Qwen2.5-VL-7B, to ViCrit-RL-7B that tops every metric. The pattern repeats at the 72B scale where ViCrit-RL-72B achieves the best performance on both ViCrit-bench and general VLM evaluation.

This finding echoes our original motivation in building ViCrit-bench, with the hypothesis that ViCrit accuracy could foreshadow a VLM's perception capability as well as the overall multimodal performance. Furthermore, the consistency of the ordering across scales suggests that ViCrit-RL's better hallucination performance in ***Finding 1*** is not a by-product of merely training on hallucination detection. Rather, models that learn to "cross-check textual claims against visual evidence" via ViCrit also perceive and reason better on general VL problems, such as chart, math, and abstract images.

### 5.3 Qualitative Results

We showcase two representative cases (Figure 6) that reveal how training on the ViCrit task sharpens visual perception and, consequently, improving VLM performance. *Example 1.* ViCrit-RL-72B is

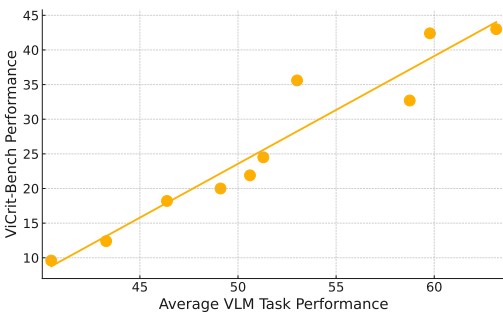

Figure 5: Correlation between average VLM task performance and ViCrit-Bench performance (Task Avg. and Overall columns in Table 2). Each point represents a different model, and the fitted linear regression line highlights a strong positive relationship, indicating that better ViCrit-Bench results are associated with higher stronger VLM capabilities.

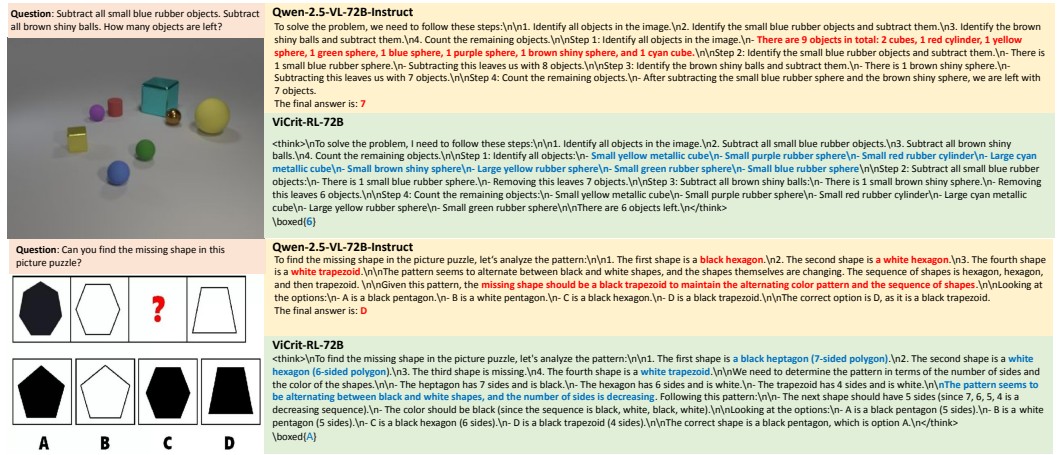

Figure 6: Two examples demonstrate the behavioral differences between models before and after training with the ViCrit task. It can be seen that ViCrit-RL-72B pays closer attention to image details and arrives at the correct final reasoning through its enhanced perception capabilities.

able to accurately identify all objects in the image in a clockwise order, capturing detailed attributes such as color and shape, and successfully making the correct calculation. *Example 2.* ViCrit-RL-72B correctly identifies all relevant visual details—including colors and the number of edges of each object—and uses this information to derive the correct answer. In contrast, Qwen2.5-VL-Instruct fails to capture the complete visual content due to its limited perception ability, leading to incorrect reasoning. These examples demonstrate that training on ViCrit task significantly improves visual perception, which is a crucial foundation for enhancing the VLM performance.

# 6  Conclusion

We have presented ViCrit, an RL proxy task that trains VLMs to pinpoint fine-grained, synthetically injected visual hallucinations in paragraph-length captions. Because each targeted span is unambiguously verifiable, ViCrit provides a challenging yet noise-free reward signal that compels models to internalize stronger perceptual strategies, yielding consistent gains across a broad suite of benchmarks. Furthermore, we release ViCrit-Bench, a carefully curated dataset that enables rigorous evaluation of VLM perception. We hope this new task will spark further breakthroughs in multimodal RL, from standalone VLMs to end-to-end-trained tool-augmented multimodal agents.

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

# Appendix

## A  Prompts used in experiments

### A.1  Prompt for Training Data Generation

We provide the prompt used for generating ViCrit task training data in Table 6.

### A.2  Prompt for ViCrit-Bench Evaluation

We provide the prompt used for ViCrit-Bench evaluation in Table 3.

Table 3: Prompt template used for ViCrit-Bench evaluation.

> **Prompt Template:**
> You are provided with an image and the description corresponding to this image. There is one hallucination in this description. Find out the hallucination phase and answer with the hallucination phase directly in a list. Your output should only be a list that contains the hallucination phase you find.
> Description:

## B  Training Hyperparameter

We provide the training hyperparameters in Table 4.

Table 4: Training Hyperparameters

| Name | Hyperparameter |
|------|----------------|
| Global batch size | 64 |
| Learning rate | $1 \times 10^{-6}$ |
| Weight decay | $1 \times 10^{-2}$ |
| KL coefficient | $1 \times 10^{-2}$ |
| Max response length | 2048 |
| Min response length | 1024 |
| Max pixels | 802816 |
| Rollout $n$ | 8 |
| Rollout batch size | 512 |
| Rollout temperature | 1.0 |
| Rollout top-$p$ | 0.99 |
| GPU memory utilization | 0.6 |

## C  Comparison with SFT

In this section, we perform SFT on Qwen-2.5-VL-7B and 72B using 900k captioning samples from PixMo-Cap, and compare the results with ViCrit-RL models trained using the same amount of data through ViCrit task RFT. As shown in Table 5, we find that although SFT significantly reduced hallucination in VLMs, it do not lead to notable performance improvements on general benchmarks—in fact, the 7B model even shows a performance drop. This highlights the effectiveness of ViCrit task RFT, which not only reduces hallucinations but also generalizes well to enhance VLM performance on general reasoning tasks.

Table 5: Comparison between ViCrit-RL and ViCrit-RL with using same captioning data for SFT. We find that although hallucinations in the VLM are significantly reduced after SFT, the performance improvement is difficult to generalize to general tasks.

| Model | Hallucination Benchmark | | | General benchamrk | | | | | | | | |
| --- | --- | --- | --- | --- | --- | --- | --- | --- | --- | --- | --- | --- |
| | CHAIRs ↓ | CHAIRi ↓ | MMHal ↑ | MathVista testmini ↑ | MathVision mini ↑ | MathVerse mini ↑ | MMMU ↑ | MMStar ↑ | MM-Vet ↑ | Blind ↑ | Charxiv reasoning ↑ | Avg. |
| Qwen2.5-VL-7B-Instruct | 28.0 | 5.1 | 3.74 | 67.8 | 23.6 | 44.5 | 50.6 | 61.7 | 66.0 | 49.3 | 41.4 | 50.61 |
| Qwen2.5-VL-7B-CapSFT | 25.5 | 4.4 | 3.78 | 67.4 | 20.1 | 44.3 | 52.1 | 53.4 | 64.7 | 47.3 | 38.0 | 48.41 |
| ViCrit-RL-7B | 25.2 | 4.5 | 3.77 | 70.7 | 25.7 | 46.3 | 52.0 | 61.9 | 67.1 | 52.6 | 47.8 | 53.01 |
| Δ (Ours - Qwen2.5-7B) | -2.8 | -0.6 | +0.03 | +2.9 | +2.1 | +1.8 | +1.4 | +0.2 | +1.1 | +3.3 | +6.4 | +2.40 |
| Qwen2.5-VL-72B-Instruct | 26.4 | 4.8 | 3.82 | 74.8 | 35.2 | 53.3 | 63.4 | 68.4 | 76.3 | 61.3 | 45.5 | 59.78 |
| Qwen2.5-VL-72B-CapSFT | 21.6 | **3.6** | 3.89 | 76.1 | 34.8 | 57.9 | 65.3 | 68.9 | 76.5 | 63.0 | 44.7 | 60.78 |
| ViCrit-RL-72B | **21.0** | 3.9 | **3.91** | **77.3** | **40.1** | **59.8** | **66.0** | **69.8** | **77.1** | **65.8** | **49.4** | **63.16** |
| Δ (Ours - Qwen2.5-72B) | -5.4 | -0.9 | +0.09 | +2.5 | +4.9 | +6.5 | +2.6 | +1.4 | +0.8 | +4.5 | +3.9 | +3.38 |

Table 6: Prompt used for training data generation.

You are a helpful assistant designed to manipulate text with precision. Your task is as follows:
1. Identify all noun phrases in a given paragraph. A noun phrase consists of a noun and its modifiers (e.g., "the wooden bridge," "a flock of birds"). Noun phrase is two to five words long. Do not output a list of multiple noun phrases.
2. Randomly select one noun phrase from the list, it can be small background objects, scene text, foreground objects. Try to select scene text and small background objects more often when possible.
3. Replace the chosen noun phrase with another phrase that is visually similar, such as changing the object attributes, replacing the object with a visually similar noun, or adding and removing characters within the scene text. The replacement should be visually similar but not identical to the original phrase. Be creative and don't always focus on the most obvious or common replacements such as color.
4. However, the replacement should introduce clear change, such that it is impossible to be ambiguous. The change should be directly related to image and be a visual description. Do not only change words to its synonyms or make ambigious changes. Do not merely change words to its synonyms. Do not merely change words to its synonyms. Do not merely change words to its synonyms.
5. Ensure the edited paragraph is still be a plausible image description, and the change is not too obvious.
6. Group the original phrase in <Before>original</Before>, and changed phrase in <After>changed</After>. <Caption> is used to give input caption and should not be generated. Perform this transformation accurately and naturally.
Here are some examples:
1. <Caption>This image appears to be a screenshot taken from an iPhone displaying the interface of a food delivery app, likely DoorDash, around the Chicago and Gary, Indiana area. The top of the screen indicates the time as 2:30 PM, with the phone connected to an LTE network. The battery icon suggests a low battery level of 15-20
Central to the image is a map highlighting various regions with color codes: red areas represent high traffic or demand, likely meaning those areas are "busy" for delivery drivers, as indicated by a red text banner. Lighter red and green sections represent varying levels of demand.
At the top of the image, a black banner labeled "Promos" is displayed, accompanied by a blue notification bell icon with the number two beside it, indicating two notifications.
The bottom of the screen shows a black navigation bar. It contains options for "Dash," "Schedule," "Account," "Ratings," and "Earnings." The "Dash" option is highlighted in red, suggesting it is currently selected. Centrally located in this bar is a red "Dash Now" button, implying that the user can begin delivering immediately. An additional black banner, located just above the navigation bar, reads "In... Hammond."
Overall, this detailed caption gives a comprehensive idea of the app's functionality, likely indicating areas of high demand where food delivery services are needed the most.</Caption>
<Before>a low battery level of 15-20%</Before>
<After>a high battery level of 75-80%</After>
2. <Caption>The image depicts a screenshot from a strategy video game with a third-person, aerial view. The central character, named Anselm, navigates through a complex, industrial-style building that evokes the aesthetic of games like Metal Gear. The environment is dark and futuristic, with certain areas illuminated, revealing various paths and stairs. The top left corner displays the yellow text "Instructor Eastwood," alongside a graph-like design. The upper center features game-related instructions in white text stating "Defensive Measures – Use Range to Your Advantage," with the notations "5" and "5.1" accompanying the instructions. Additionally, a map of the area is situated in the lower left corner, while the bottom right corner features interactive elements or a key potentially indicating available weapons. The overall scene suggests a mission-focused gameplay scenario requiring strategic maneuvering and tactical decision-making.</Caption>
<Before>text "Instructor Eastwood,"</Before>
<After>text "Instructor Westwood,"</After>
3. <Caption>The image depicts a three-dimensional panoramic view of a conference room where a business meeting is taking place. The setting is a typical meeting room with white walls, fluorescent lighting, and windows equipped with blinds on some, including wooden slats on one. At the center of the room is a round, yellow table that appears slightly distorted due to the panoramic effect. On this table, there are various items, including a box of tissues, a white mug, a teacup, and pamphlets.
Surrounding the table, seated in a circle, are eight individuals. They appear to be a mixed group of men and women, predominantly of Asian descent, and are dressed in a variety of attire ranging from business to casual. All attendees are wearing name tags on the left side of their chests, indicating their participation in the meeting. Their seating arrangement includes black chairs with green backs, and each person either has their hands folded in their laps or is holding something, possibly a drink.
From left to right, the attendees include a woman in a peach-colored t-shirt with writing, a man in a blue shirt, a woman in a gray sweater, a woman in a green shirt, another woman in a blazer, an empty chair, a man in a yellowish shirt, a man wearing a ball cap, and a man in a blue and green jacket. One notable aspect is that the meeting environment, though professional, is quite understated with minimalistic decor and standard conference room furnishings.</Caption>
<Before>eight individuals</Before>
<After>seven individuals</After>
Here is the input caption: <Caption>{**CAPTION**}</Caption>

