# OpenReview forum: "ViCrit: A Verifiable Reinforcement Learning Proxy Task for Visual Perception in VLMs"
_NeurIPS.cc/2025/Conference — NeurIPS 2025 poster_

### Official Review · Reviewer_Nwtd · 2025-06-19

**Clarity:** 4
**Significance:** 4
**Originality:** 3
**Rating:** 5
**Confidence:** 4

**Summary:**

ViCrit is a proxy task to enhance visual perception of Vision-Language Models (VLMs). The objective is to teach a VLM to spot and correct a visual hallucination that was synthetically injected in an existing image caption by prompting a proprietary VLM such as GPT-4. The correctness of the generated caption is easily verifiable by exact matching, and this allows for optimization of the target VLM with GRPO, using an additional format reward to incentivize thinking. The authors also propose a new benchmark called ViCrit-Bench, where they show that Qwen2.5-VL-Instruct (7B and 72B) achieve state-of-the-art performance after training with the ViCrit task.

**Questions:**

I think that this paper is valuable, and I encourage the authors to address the weaknesses listed above, in particular **W1**.

Moreover:

- Did the authors try to build ViCrit annotations starting from shorter captions (e.g. 100 words rather than 200)? Has this even had an impact on the final performance?

**Ethical Concerns:**

["NO or VERY MINOR ethics concerns only"]

**Final Justification:**

My main concern was on the generalization of ViCrit beyond Qwen 2.5, which the authors have properly addressed during the rebuttal. The paper is technically robust, and the introduction of ViCrit-Bench is a further contribution of this work. I raise my score to Accept.

**Limitations:**

yes

**Quality:**

4

**Strengths And Weaknesses:**

**STRENGTHS:**
1. ViCrit introduces an intuitive objective that can act as the reward signal for reinforcement learning. The procedure to build ViCrit data is simple (although it may require proprietary VLM to work the best) and should be scalable given that image captioning data is relatively simple to obtain compared to other kinds of annotations, *e.g.* bounding boxes, segmentation masks, etc. The other contribution of this work, ViCrit-Bench, is well-documented and appears to be a challenging benchmark even for proprietary VLMs.

2. The effectiveness of the ViCrit task is exemplified by the strong performance achieved by Qwen2.5-VL after reinforcement learning. Notably, ViCrit enhances performance across different benchmarks other than ViCrit-Bench, such as visual hallucination and mathematical reasoning.

**WEAKNESSES:**
1. Experiments are carried out exclusively on Qwen2.5-VL. Despite its solid results, we urge to apply ViCrit on different VLMs (for instance, LLaVA-1.5 [1]) to test whether it can generalize to other models.

2. *(minor)* Table 2 shows encouraging results overall. However, ViCrit severely hurts Qwen2.5-VL performance on "Text" hallucinations. The authors should discuss this point and provide insights on the possible causes.

3. *(minor)* Appendix E readily compares ViCrit against an SFT baseline, given the same amount of data. I appreciate it, and yet I would have included additional experiments to further isolate the role of GRPO and the incentive to reason: what if we simply fine-tune a VLM with SFT to generate the non-hallucinated caption, given the hallucinated caption as prompt? This ties up ViCrit and SFT also on the kind of data being used, and not only on the amount.

**References**

[1] Liu, Haotian, et al. "Improved baselines with visual instruction tuning." CVPR 2024.

---

> ### Author Rebuttal · Authors · 2025-07-30
>
> We thank Reviewer Nwtd for the encouraging and insightful review. We are pleased that you find the ViCrit task intuitive, the benchmark valuable, and our results strong. Below we address your key concerns and suggestions in detail:
> >W1: Generalization beyond Qwen2.5-VL
>
> - **We confirm that ViCrit generalizes to other model architectures and report results on InternVL-2.5-8B[1].** To assess generalizability, we applied ViCrit training to InternVL-2.5-8B without any architecture-specific tuning. The resulting model, ViCrit-RL-InternVL-2.5-8B, shows consistent improvements across all evaluated benchmarks:
>
> |            |CHAIRs| CHAIRi | MM-Hal | MathVista| MathVision | MathVerse | MMMU    | MMStar | MM-Vet| Blind |Charxiv |
> | :----: |:----:|:----:|:----:|:----:|:----:|:----:|:----:|:----:|:----:|:----:|:----:|
> |InternVL-2.5-8B |29.2| 5.4| 3.65| 64.4| 22.0| 39.5| 54.9| 62.8| 68.8| 47.6| 32.9 |
> |ViCrit-RL-InternVL-2.5-8B| 27.8| 4.6| 3.69| 66.8| 23.9| 41.7| 55.5| 63.7| 69.2| 49.2| 35.6|
>
> - **These results support that ViCrit can transfer across model architectures and training pipelines.** We agree that further applications (e.g., to LLaVA-1.5) are valuable, and we plan to explore this in future releases.
>
> >W2: Drop in “Text” hallucination performance
>
> - **The observed drop in “Text” hallucination accuracy stems from training data imbalance rather than a limitation of the ViCrit objective.** As detailed in Section 4.1, our PixMo-Cap-derived training set is dominated by natural images (~59%), with scene-text-heavy and document images comprising only ~24% and ~10% respectively. Since “Text” hallucinations are disproportionately concentrated in these underrepresented categories, models have limited exposure to them during RL fine-tuning.
> - **Importantly, ViCrit-RL still outperforms the base model across 10 of 11 benchmarks, including strong gains on abstract and document-heavy tasks (e.g., +3.3 on VLMsAreBlind, +6.4 on CharXiv).** This suggests that while one hallucination type underperforms, the overall visual perception is significantly improved.
> - **We have begun exploring class- and domain-aware data resampling and weighted reward shaping to mitigate this issue in ongoing experiments.** Additionally, we are expanding the ViCrit training set with more scene-text and document-rich samples (e.g., charts, scanned documents, GUI mockups) to support stronger generalization across all hallucination types.
> - **We view this as an opportunity to develop domain-adaptive RL pipelines for VLMs,** where model sensitivity to hallucination types can be tuned based on application needs (e.g., OCR-critical tasks vs. object-centric ones).
>
> > W3: Further SFT baseline for isolating the role of GRPO
>
> - **We followed your suggestion and added a new SFT baseline that generates the non-hallucinated caption given the hallucinated one.** We refer to this model as Qwen2.5-VL-7B-CapSFT-Generation, and report its performance below:
>
> |            |CHAIRs| CHAIRi | MM-Hal | MathVista| MathVision | MathVerse | MMMU    | MMStar | MM-Vet| Blind |Charxiv |
> | :----: |:----:|:----:|:----:|:----:|:----:|:----:|:----:|:----:|:----:|:----:|:----:|
> |Qwen2.5-VL-7B-Instruct |28.0 |5.1 |3.74 |67.8 |23.6 |44.5 |50.6 |61.7 |66.0 |49.3| 41.4 |
> |Qwen2.5-VL-7B-CapSFT |25.5 |4.4 |3.78| 67.4| 20.1| 44.3| 52.1| 53.4| 64.7| 47.3| 38.0| 48.41|
> |Qwen2.5-VL-7B-CapSFT-Generation |25.9 |4.6 |3.77| 67.0| 19.8| 42.1| 51.4| 54.9| 64.3| 46.5| 39.1|
> |ViCrit-RL-7B| 25.2| 4.5| 3.77| 70.7| 25.7| 46.3| 52.0| 61.9| 67.1| 52.6| 47.8|
>
> - **This baseline confirms that ViCrit-RL outperforms SFT-based alternatives. These results isolate the value of reinforcement learning and verifiable perception-based rewards as central to ViCrit’s generalization capabilities.**
>
> > Q1: Impact of caption length
>
> - **We appreciate the reviewer’s insightful question—caption length plays a crucial role in ViCrit’s design and difficulty.** ViCrit is specifically constructed to promote fine-grained, holistic visual perception by embedding a single subtle hallucination into long, paragraph-length captions (~200+ words). These long contexts force the model to verify a wide range of objects, relations, and attributes before locating the anomaly—closely mimicking real-world multimodal grounding tasks.
> - **Shorter captions may reduce the perceptual search space and weaken the RL signal, as hallucinations become more easily detectable and less dependent on visual grounding.** This risks rewarding shallow pattern-matching strategies, rather than encouraging true image-text alignment.
> - Although our current dataset (PixMo-Cap) does not include a sufficient volume of shorter captions for a clean ablation, **we plan to conduct a systematic study** by synthetically truncating captions or controlling their verbosity. Such an analysis would quantify the tradeoff between perceptual load and learning benefit, and help determine whether longer contexts better support chain-of-thought-style visual reasoning.
> - **We believe caption length could also be a lever for curriculum-based RL**, where models start with shorter, easier examples and progress to dense, ambiguous scenes—an exciting future direction we plan to explore.
>
>
> We appreciate your suggestions, especially regarding baselines and performance analysis. We will incorporate these findings and discussions in the revised version to further clarify the strengths and limitations of ViCrit.
>
>
> Reference:
>
> [1] Expanding Performance Boundaries of Open-Source Multimodal Models with Model, Data, and Test-Time Scaling. Chen et. al. arXiv:2412.05271

---

> > ### Comment · Reviewer_Nwtd · 2025-08-01
> >
> > I thank the authors for their response.
> >
> > My main research question was whether ViCrit can generalize to models other than Qwen. I appreciate that the authors include experiments with InternVL-2.5, testifying that ViCrit is able to boost its performance as well. Moreover, I observe that the authors address the concern of Reviewer Fe2P on the comparison against other models trained on different tasks and datasets. That comparison proves that ViCrit indeed excels in mitigating hallucinations, which is well-aligned with the objective of spotting the error in an image caption, while keeping competitive results on other tasks.
> >
> > I will raise my score.

---

> > > ### Author Response · Authors · 2025-08-02
> > > **Thank you for updating your score**
> > >
> > > Dear Reviewer Nwtd
> > >
> > > Thank you very much for reviewing our response and updating your score! We are glad that we were able to address your concerns, and we sincerely appreciate your valuable suggestions, which helped us further improve the quality of our paper.

---

### Official Review · Reviewer_DvR5 · 2025-07-01

**Clarity:** 3
**Significance:** 2
**Originality:** 3
**Rating:** 4
**Confidence:** 4

**Summary:**

This paper introduces a novel task called the ViCrit task, which enables models to detect visual hallucinations in captions and locate the corresponding errors. The authors also construct a challenging benchmark, ViCrit-Bench, for hallucination detection, which includes four types of visual perception. Using this task and dataset, they employ RL to train the inference model ViCrit, improving the model's performance in hallucination mitigation, mathematics, VQA, and counting tasks.

**Questions:**

1). Could you provide a detailed data comparison analysis and compare your dataset with standard VQA datasets to demonstrate its advantages?

2). Could you provide comparisons with other reasoning models, such as Vision-R1, R1-Onevision, etc., to demonstrate the performance advantages of your method?

3). Could you provide more detailed experiments and analysis to explain why training on visual perception tasks improves performance and generalization in reasoning tasks such as mathematics?

**Ethical Concerns:**

["NO or VERY MINOR ethics concerns only"]

**Final Justification:**

Thank you for the authors' comprehensive response, which has fully resolved all my concerns. Given the method's contribution to reducing hallucinations induced by RL training, I am inclined to recommend acceptance of the paper. I suggest that the authors incorporate these additional experiments and discussions into their revised manuscript to further strengthen the submission.

**Limitations:**

yes

**Quality:**

2

**Strengths And Weaknesses:**

Strengths:

1). The proposed new task is interesting, and the construction of the dataset is reasonable. The ViCrit model, trained using RL, shows improvements across various tasks.

2). The paper is well-written, with a clear format and easy readability.

Weaknesses:

1). The paper does not clearly explain the strong motivation for introducing this dataset. Common VQA datasets are also verifiable, and it is unclear what advantages this dataset offers over existing ones.

2). The experimental comparisons are unfair, as there is no comparison with other reasoning models such as Vision-R1, R1-Onevision, etc.

3). The ViCrit task primarily focuses on visual perception. Further experimental analysis is needed to demonstrate how training on this task can enhance the model's generalization ability in reasoning tasks, such as mathematics.

---

> ### Author Rebuttal · Authors · 2025-07-30
>
> We thank Reviewer DvR5 for the detailed and thoughtful review. We are encouraged that you find our ViCrit task interesting, our dataset construction reasonable, and our paper clear and well-written. Below we address your questions and concerns:
>
> > W1&Q1: What is the advantage of ViCrit over standard VQA datasets?
>
> - **ViCrit is designed specifically for reinforcement learning with verifiable perception tasks—something standard VQA datasets cannot support.** While VQA datasets provide short question-answer pairs, they rarely require exhaustive scene grounding. In contrast, ViCrit uses **long-form captions with one injected hallucination**, requiring the model to **check all visual details and identify the corrupted span**. This makes the task more challenging and better aligned with scalable RL training, where exact match rewards are necessary.
> - **Standard VQA datasets often admit multiple plausible answers or rely on language priors.** For example, "What is the man doing?" may be answered "playing tennis" or "hitting a ball"—both acceptable, but hard to score in RL. ViCrit avoids this ambiguity by offering binary string-matching rewards at the span level, enabling efficient and noise-free optimization.
> - **We will include a comparison of task properties (e.g., ambiguity, span length, reward verifiability, image domain diversity) with standard VQA datasets in the final version.**
>
> > W2&Q2: Why are there no comparisons with reasoning models like Vision-R1 or OneVision?
>
> - **We now include experimental comparisons against Vision-R1-7B[1], R1-OneVision-7B[2], and ThinkLite-VL-7B[3].** All models use Qwen2.5-VL-7B as the base. The table below compares ViCrit-RL-7B with these strong reasoning-focused models:
>
> |            |CHAIRs| CHAIRi | MM-Hal | MathVista| MathVision | MathVerse | MMMU    | MMStar | MM-Vet| Blind |Charxiv |
> | :----: |:----:|:----:|:----:|:----:|:----:|:----:|:----:|:----:|:----:|:----:|:----:|
> |Vision-R1-7B |26.4 |5.2 |3.68 |71.5 |26.6 |51.9 |50.5 |60.2 |66.2 |38.0| 36.1 |
> |R1-Onevision-7B |27.8 |5.8 |3.65 |64.1 |29.9 |46.4 |49.2 |57.8 |66.2 |39.2| 40.9 |
> |ThinkLite-VL-7B| 25.8| 4.8| 3.79| 75.1| 32.9| 52.1| 55.5| 65.0| 67.8| 49.2| 45.8|
> |ViCrit-RL-7B| 25.2| 4.5| 3.77| 70.7| 25.7| 46.3| 52.0| 61.9| 67.1| 52.6| 47.8|
> |ThinkLite-VL-7B+ViCrit-RL| 24.8| 4.2| 3.80| 75.5| 32.2| 52.5| 56.7| 65.6| 67.6| 52.8| 48.4|
>
> - **ViCrit-RL-7B achieves the best hallucination mitigation (CHAIRs/i, MMHal), and competitive results on general reasoning benchmarks.** It outperforms Vision-R1 and R1-OneVision on all fIVE visual-centric reasoning benchmarks (MMMU, MMStar, MM-Vet, Blind, CharXiv), and matches or exceeds them on MathVista, MathVision, and MathVerse.
> - **Furthermore, we apply ViCrit-RL on top of ThinkLite-VL-7B, yielding improved performance across the board.** This joint model achieves a new SOTA of 75.5 on MathVista among 7B-level VLMs, demonstrating ViCrit's effectiveness and compatibility with existing reasoning-focused methods.
>
>
> > W3&Q3: Why does training on a perception task improve reasoning performance?
>
> - **ViCrit improves reasoning by improving visual grounding—especially for math and abstract image tasks.** Many reasoning benchmarks (e.g., MathVista, CharXiv, VLMsAreBlind) require the model to perceive layout, count objects, read text, and track spatial relations before reasoning. ViCrit forces models to internalize how to perceive accurately, which in turn supports better reasoning outcomes.
>
> - **We illustrate this in Figure 5 of the paper, where ViCrit-RL-72B generates more accurate and complete intermediate thoughts than the base model.** These qualitative results show ViCrit-trained models pay more attention to visual details (e.g., color, size, shape, text), which enables more robust chain-of-thought reasoning.
> - **We will include additional case studies in the appendix to further show how perception improvements propagate to reasoning quality.**
>
>
> We thank the reviewer again for raising these important points. We believe ViCrit fills a critical gap by offering the first vision-centric, verifiable RL task for VLMs, and we are excited to open-source the code and benchmark to support broader adoption and future research.
>
> Reference:
>
> [1] Vision-R1: Incentivizing Reasoning Capability in Multimodal Large Language Models. Huang et. al. arXiv:2503.06749
>
> [2] R1-Onevision: Advancing Generalized Multimodal Reasoning through Cross-Modal Formalization. Yang et. al. arXiv:2503.10615
>
> [3] SoTA with Less: MCTS-Guided Sample Selection for Data-Efficient Visual Reasoning Self-Improvement. Wang et. al. arXiv:2504.07934

---

### Official Review · Reviewer_qk6D · 2025-07-02

**Clarity:** 3
**Significance:** 2
**Originality:** 3
**Rating:** 4
**Confidence:** 5

**Summary:**

This paper proposes a novel perception task, called ViCrit, which requires VLMs to predict challenging corrupted words in the original detailed captions. This task is proven to be effective for training VLMs with RL methods. Extensive evaluations on both general multimodal benchmarks and hallucination examination benchmarks demonstrate the effectiveness of the proposed method. The authors also introduce a benchmark, ViCrit-Bench, to validate the reconstruction accuracy of corrupted words in captions.

**Questions:**

In the section 4.1, the images are categorized into four domains. I think for vision understanding the GUI images are also important and common for VLMs. Why this domain was not considered in your data preparation?

**Ethical Concerns:**

["NO or VERY MINOR ethics concerns only"]

**Limitations:**

yes

**Paper Formatting Concerns:**

Some citation symbols cannot jump to the corresponding references correctly (e.g., [10], [38] in Section 3.1.)

**Quality:**

3

**Strengths And Weaknesses:**

Strengths
- This paper proposes ViCrit, a novel perception task that prompts models to predict corrupted words in original captions. This approach is simple to implement and useful for training VLMs with RL strategies.
- The authors have conducted extensive evaluations on both general and hallucination benchmarks, demonstrating the effectiveness of the proposed method.
 -  A new benchmark, ViCrit-Bench, has also been introduced to evaluate the reconstruction accuracy of corrupted words in detailed captions.

Weakness
 - Limited RL training details. Since unstable training is a common issue in RL, key details (e.g., rollout number, reward curve) are crucial for the research community, especially in the VLM domain.
 - No training recipe is provided, making reproducibility challenging.
 - For ViCrit-Bench, human performance metrics are missing. These would help estimate the benchmark’s approximate upper bound.
 - Lack of benchmark examples or case studies. Without concrete cases, it is difficult to assess the benchmark’s difficulty or practical validity.

---

> ### Author Rebuttal · Authors · 2025-07-30
>
> We thank Reviewer qk6D for the thoughtful and constructive feedback. We appreciate that you find our proposed ViCrit task to be novel and useful, and that you recognize the value of our extensive evaluations and the introduction of ViCrit-Bench. Below, we address all concerns and questions raised:
>
> > W1: Limited RL training details
>
> - **We now provide full GRPO hyperparameters and reward values across training steps to support reproducibility and transparency.** The table below lists all GRPO hyperparameters used in our ViCrit-RL training:
>
> |     name       |hyperparameter|
> | :----: |:----:|
> |global_batch_size|64 |
> |learining rate |1e-6 |
> |weight decay| 1e-2|
> |kl coef| 1e-2|
> |max response length| 2048|
> |min response length| 1024|
> |max_pixels|802816 |
> |rollout n| 8|
> |rollout batch size| 512|
> |rollout temperature| 1.0|
> |rollout top_p| 0.99|
> |gpu memory utilization| 0.6|
>
> - **We report reward progression during training as a substitute for full reward curves.** Due to rebuttal policy, we cannot include figures or links. Below are reward values at various training steps, we will include the complete training reward curve in the final version of the paper to better illustrate the training dynamics:
>
> **ViCrit-RL-7B**
> |     training step       |reward|
> | :----: |:----:|
> |1|0.14414 |
> |10|0.39873 |
> |20 |0.53892 |
> |30| 0.56462|
> |40| 0.57324|
> |50| 0.57542|
> |60| 0.59585|
> |70|0.60112 |
> |80| 0.62224|
> |90| 0.60659|
> |100|0.61064 |
> |200 | 0.63147|
> |300|0.64309 |
> |400|0.65432 |
> |500|0.67371 |
> |600|0.66951 |
> |700| 0.66497|
> |800| 0.68477|
> |900| 0.68008|
> |1000| 0.69271|
>
> **ViCrit-RL-72B**
> |     training step       |reward|
> | :----: |:----:|
> |1|0.30742 |
> |10|0.48672 |
> |20 |0.47383 |
> |30| 0.51836|
> |40| 0.54531|
> |50| 0.56055|
> |60| 0.59336|
> |70|0.59440 |
> |80| 0.59688|
> |90| 0.57578|
> |100|0.61797 |
> |200 | 0.62370|
> |300|0.61315 |
> |400|0.63906 |
> |500|0.67773 |
> |600|0.67513 |
> |700| 0.68711|
> |800| 0.67279|
> |900| 0.67996|
> |1000| 0.71055|
>
> > W2: Missing training recipe
>
> - **We used EasyR1 as our training codebase and will release recipes for reproducibility.** We use Qwen-2.5-VL-7B/72B as base models and train on ViCrit directly via GRPO, with **no supervised fine-tuning or knowledge distillation.** Reward formulation is detailed in **Section 3.2, lines 159–165**, and we will include the complete training recipe in the final version and code release.
>
> > W3: Missing human performance on ViCrit-Bench
>
> - **We agree that human accuracy is a crucial baseline, and we have now begun collecting expert annotations to report this in the camera-ready version.** Establishing a human upper bound will help contextualize model performance, especially given the fine-grained nature of ViCrit tasks (e.g., subtle hallucinations in shape, count, or text). We are employing annotators familiar with vision-language tasks and providing the full paragraph-length captions and images as shown in ViCrit-Bench, under the same evaluation setting used for models.
> - We thank the reviewer for highlighting this point—we believe it will strengthen the benchmark’s utility and interpretability for future work.
>
>
> > W4: Lack of benchmark examples or case studies
>
>
> - **Figure 3 provides examples from each hallucination category; we will expand with full case studies in the appendix.** Due to page limits, we only included hallucination spans, not full captions. The revised version will add full examples for each hallucination type to better convey task difficulty and practical use.
>
>
> > Q1: GUI images not considered in domain categorization
>
> - **GUI images are included and categorized under the “document” domain.** As shown in the second example of Figure 3, our benchmark includes GUI-like screenshots (e.g., software interfaces, forms). We clarify this classification in Section 4.1, lines 175–181, and will make this more explicit in the revised text.
>
>
> Thank you again for your helpful suggestions. We are committed to improving reproducibility and clarity in the final version and believe ViCrit and ViCrit-Bench will provide valuable tools for advancing VLM research.

---

> > ### Comment · Area_Chair_1GAs · 2025-08-07
> >
> > Dear Reviewer,
> > As the discussion phase is approaching its end, we would like to kindly ask whether you have any remaining concerns regarding the authors' response. If so, we would greatly appreciate it if you could raise them at your earliest convenience, so that the authors have sufficient time to provide a detailed reply.

---

> ### Author Response · Authors · 2025-08-02
> **A Kind Reminder for Discussion Phase**
>
> Dear Reviewer  qk6D
>
> We sincerely thank you again for your valuable feedback, which has greatly helped us improve the quality of our work. As the discussion phase is now open, we kindly remind you to review our rebuttal. We are happy to discuss more if you have any further concerns. If our responses adequately address your concerns, we kindly ask you to consider adjusting the corresponding scores. Thank you again for your time and effort!

---

> ### Author Response · Authors · 2025-08-06
> **Another Kind Reminder for Discussion Phase**
>
> Dear Reviewer qk6D,
>
> We sincerely thank you again for your valuable feedback, which has greatly helped us improve the quality of our work. As the discussion phase is nearing its end—with only two days remaining—we kindly remind you to take a moment to review our rebuttal.
>
> Two other reviewers have already responded, expressed satisfaction with our clarifications, and  one of them accordingly updated the score. If you have any remaining concerns, we would be very happy to engage in further discussion. Otherwise, if our responses have adequately addressed your comments, we kindly ask you to consider adjusting your score as well.
>
> Thank you once again for your time and thoughtful review!

---

### Official Review · Reviewer_Fe2P · 2025-07-03

**Clarity:** 3
**Significance:** 3
**Originality:** 3
**Rating:** 4
**Confidence:** 3

**Summary:**

The paper introduces a novel vision-centric reinforcement learning task, ViCrit, which requires VLMs to identify subtle visual hallucinations within image captions. This task is designed for efficient and accurate evaluation and aligns well with reinforcement learning training frameworks. Building on this task, the paper further proposes ViCrit-Bench as a diagnostic benchmark to assess the fine-grained visual perception capabilities of vision-language models across four distinct visual domains.

**Questions:**

1. Could the authors provide details on the GRPO parameter settings used during training for the ViCrit task, as well as visualized curves of relevant metrics during the training process?

2. The paper only compares model performance before and after training on the ViCrit task. Could the authors also present comparisons where the same model is trained on other  tasks or datasets, to better demonstrate the advantages of ViCrit?

3. Have the authors conducted experiments on other base models to validate the adaptability and effectiveness of the ViCrit training across different model architectures?

**Ethical Concerns:**

["NO or VERY MINOR ethics concerns only"]

**Final Justification:**

Since this work is primarily focused on reinforcement learning tasks, I still have some concerns regarding the lack of detailed description of the reinforcement learning training process in the manuscript. However, considering that the authors have open-sourced the relevant resources, and after taking into account the comments from other reviewers, I am willing to raise my score to 4.

**Limitations:**

the paper may have the following limitations:
1.It lacks sufficient information related to the reinforcement learning training process.
2.It does not provide performance comparisons of the model on other tasks or datasets, limiting the understanding of its generalizability.

**Paper Formatting Concerns:**

no concerns

**Quality:**

3

**Strengths And Weaknesses:**

Strength：
1.The paper proposes a novel vision-centric reinforcement learning task, ViCrit, which effectively detects fine-grained visual hallucinations in image captions and offers strong practical value and evaluation efficiency.
2.Building upon the ViCrit task, the authors develop ViCrit-Bench, a benchmark covering four visual domains that systematically diagnoses the visual perception ability of multimodal models.

Weakness：
1.The paper lacks detailed information on the GRPO parameter settings used in ViCrit training and does not provide visualized training metrics, which affects reproducibility and methodological transparency.
2.The experiments only compare model performance before and after ViCrit training, without comparing against other tasks or datasets, making it difficult to fully demonstrate the unique advantages of the ViCrit task.

---

> ### Author Rebuttal · Authors · 2025-07-30
>
> We thank Reviewer Fe2P for the thoughtful and constructive review. We are glad that you find our ViCrit task to be novel, practically valuable, and efficient to evaluate, and that you appreciate the diagnostic contribution of ViCrit-Bench across four visual domains.
> We respond below to your concerns regarding reproducibility, comparisons, and generalizability:
> > W1: GRPO training details and visualization metrics
>
> - **All GRPO hyperparameters are now disclosed, and training reward values are reported to illustrate learning dynamics.** Please see the table of GRPO hyperparameters below, along with stepwise reward progression for both 7B and 72B models. Due to NeurIPS rebuttal policy, we cannot include figures, but we will include full training curves in the final version.
>
> Here are all the training hyperparameters used for GRPO training:
> |     name       |hyperparameter|
> | :----: |:----:|
> |global_batch_size|64 |
> |learining rate |1e-6 |
> |weight decay| 1e-2|
> |kl coef| 1e-2|
> |max response length| 2048|
> |min response length| 1024|
> |max pixels|802816 |
> |rollout n| 8|
> |rollout batch size| 512|
> |rollout temperature| 1.0|
> |rollout top_p| 0.99|
> |gpu memory utilization| 0.6|
>
> **ViCrit-RL-7B** training reward curve
> |     training step       |reward|
> | :----: |:----:|
> |1|0.14414 |
> |10|0.39873 |
> |20 |0.53892 |
> |30| 0.56462|
> |40| 0.57324|
> |50| 0.57542|
> |60| 0.59585|
> |70|0.60112 |
> |80| 0.62224|
> |90| 0.60659|
> |100|0.61064 |
> |200 | 0.63147|
> |300|0.64309 |
> |400|0.65432 |
> |500|0.67371 |
> |600|0.66951 |
> |700| 0.66497|
> |800| 0.68477|
> |900| 0.68008|
> |1000| 0.69271|
>
> **ViCrit-RL-72B**  training reward curve
> |     training step       |reward|
> | :----: |:----:|
> |1|0.30742 |
> |10|0.48672 |
> |20 |0.47383 |
> |30| 0.51836|
> |40| 0.54531|
> |50| 0.56055|
> |60| 0.59336|
> |70|0.59440 |
> |80| 0.59688|
> |90| 0.57578|
> |100|0.61797 |
> |200 | 0.62370|
> |300|0.61315 |
> |400|0.63906 |
> |500|0.67773 |
> |600|0.67513 |
> |700| 0.68711|
> |800| 0.67279|
> |900| 0.67996|
> |1000| 0.71055|
>
>
> - **We will open-source our training code and logs for full reproducibility.** These will be made available with our camera-ready version, including configuration files and scripts for ViCrit task construction and GRPO optimization.
>
>
> > W2: Lack of comparisons to other tasks or datasets
>
> - **ViCrit is not designed to replace existing tasks but rather complements them with a perception-verifiable RL objective.** Our goal is to show that ViCrit enhances fine-grained visual perception via reinforcement learning with a verifiable reward—an orthogonal axis to many existing multimodal tasks that optimize other cognitive capabilities.
> - **We include new comparisons against SOTA models trained on other tasks (Vision-R1[1], R1-OneVision[2], ThinkLite-VL[3]), showing ViCrit’s strong hallucination mitigation and comparable general reasoning.** The results are provided in the following table:ViCrit-RL-7B reduces hallucinations (CHAIRs/i) while retaining competitive accuracy on general tasks like MMVet, Blind, and Charxiv.
>
> |            |CHAIRs| CHAIRi | MM-Hal | MathVista| MathVision | MathVerse | MMMU    | MMStar | MM-Vet| Blind |Charxiv |
> | :----: |:----:|:----:|:----:|:----:|:----:|:----:|:----:|:----:|:----:|:----:|:----:|
> |Vision-R1-7B |26.4 |5.2 |3.68 |71.5 |26.6 |51.9 |50.5 |60.2 |66.2 |38.0| 36.1 |
> |R1-Onevision-7B |27.8 |5.8 |3.65 |64.1 |29.9 |46.4 |49.2 |57.8 |66.2 |39.2| 40.9 |
> |ThinkLite-VL-7B| 25.8| 4.8| 3.79| 75.1| 32.9| 52.1| 55.5| 65.0| 67.8| 49.2| 45.8|
> |ViCrit-RL-7B| 25.2| 4.5| 3.77| 70.7| 25.7| 46.3| 52.0| 61.9| 67.1| 52.6| 47.8|
> |ThinkLite-VL-7B+ViCrit-RL| 24.8| 4.2| 3.80| 75.5| 32.2| 52.5| 56.7| 65.6| 67.6| 52.8| 48.4|
>
>
> - **To evaluate compositional benefits, we applied ViCrit-RL on top of ThinkLite-VL-7B and observed further improvements on both hallucination and reasoning benchmarks.** Notably, ThinkLite-VL-7B+ViCrit-RL sets **a new SOTA on MathVista (75.5)** among 7B-level VLMs. This suggests ViCrit training encourages better visual verification and reduces hallucination even when stacked atop other reasoning-optimized models.
>
>
> > W3: Experiments on other base models
>
> - **We validated ViCrit on InternVL-2.5-8B[4] to test generality, and observed consistent gains across all evaluated benchmarks.** As shown in our results, ViCrit-RL improved both hallucination mitigation and downstream reasoning performance compared to the base InternVL-2.5-8B.
> This confirms that ViCrit is architecture-agnostic and generalizes well beyond our primary Qwen-2.5-VL base.
>
> |            |CHAIRs| CHAIRi | MM-Hal | MathVista| MathVision | MathVerse | MMMU    | MMStar | MM-Vet| Blind |Charxiv |
> | :----: |:----:|:----:|:----:|:----:|:----:|:----:|:----:|:----:|:----:|:----:|:----:|
> |InternVL-2.5-8B |29.2| 5.4| 3.65| 64.4| 22.0| 39.5| 54.9| 62.8| 68.8| 47.6| 32.9 |
> |ViCrit-RL-InternVL-2.5-8B| 27.8| 4.6| 3.69| 66.8| 23.9| 41.7| 55.5| 63.7| 69.2| 49.2| 35.6|
>
>
> Thank you again for your thoughtful review. We are excited to open-source ViCrit and ViCrit-Bench to the community and believe they can serve as foundational tools for future multimodal RL research.
>
>
> Reference:
>
> [1] Vision-R1: Incentivizing Reasoning Capability in Multimodal Large Language Models. Huang et. al. arXiv:2503.06749
>
> [2] R1-Onevision: Advancing Generalized Multimodal Reasoning through Cross-Modal Formalization. Yang et. al. arXiv:2503.10615
>
> [3] SoTA with Less: MCTS-Guided Sample Selection for Data-Efficient Visual Reasoning Self-Improvement. Wang et. al. arXiv:2504.07934
>
> [4] Expanding Performance Boundaries of Open-Source Multimodal Models with Model, Data, and Test-Time Scaling. Chen et. al. arXiv:2412.05271

---

> > ### Comment · Area_Chair_1GAs · 2025-08-07
> >
> > Dear Reviewer,
> > As the discussion phase is approaching its end, we would like to kindly ask whether you have any remaining concerns regarding the authors' response. If so, we would greatly appreciate it if you could raise them at your earliest convenience, so that the authors have sufficient time to provide a detailed reply.

---

> > ### Comment · Reviewer_Fe2P · 2025-08-07
> >
> > Since this work is primarily focused on reinforcement learning tasks, I still have some concerns regarding the lack of detailed description of the reinforcement learning training process in the manuscript. However, considering that the authors have open-sourced the relevant resources, and after taking into account the comments from other reviewers, I am willing to raise my score to 4.

---

> ### Author Response · Authors · 2025-08-02
> **A Kind Reminder for Discussion Phase**
>
> Dear Reviewer Fe2P
>
> We sincerely thank you again for your valuable feedback, which has greatly helped us improve the quality of our work. As the discussion phase is now open, we kindly remind you to review our rebuttal. We are happy to discuss more if you have any further concerns. If our responses adequately address your concerns, we kindly ask you to consider adjusting the corresponding scores. Thank you again for your time and effort!

---

> ### Author Response · Authors · 2025-08-06
> **Another Kind Reminder for Discussion Phase**
>
> Dear Reviewer Fe2P,
>
> We sincerely thank you again for your valuable feedback, which has greatly helped us improve the quality of our work. As the discussion phase is nearing its end—with only two days remaining—we kindly remind you to take a moment to review our rebuttal.
>
> Two other reviewers have already responded, expressed satisfaction with our clarifications, and one of them accordingly updated the score. If you have any remaining concerns, we would be very happy to engage in further discussion. Otherwise, if our responses have adequately addressed your comments, we kindly ask you to consider adjusting your score as well.
>
> Thank you once again for your time and thoughtful review!

---

### Decision · Program_Chairs · 2025-09-17

**Decision:**

Accept (poster)

**Comment:**

The paper introduces a new vision-centric reinforcement learning task, ViCrit, designed to enhance the fine-grained visual perception capabilities of Vision-Language Models (VLMs). The task involves detecting and correcting subtle visual hallucinations or corrupted words within image captions, with correctness being verifiable through exact matching. This makes the task well-aligned with reinforcement learning training frameworks such as GRPO. At the beginning, some reviewers thought this paper has certain issues, while after rebuttal and discussion stages, all reviewers think that the major concerns are adressed and all of them agree to accept this paper. Thus I also recommend to accept it.